# Challenging Patterns of Atypical Dermatofibromas and Promising Diagnostic Tools for Differential Diagnosis of Malignant Lesions

**DOI:** 10.3390/diagnostics13040671

**Published:** 2023-02-10

**Authors:** Olguța Anca Orzan, Alexandra Maria Dorobanțu, Cristian Dorin Gurău, Sibel Ali, Mara Mădălina Mihai, Liliana Gabriela Popa, Călin Giurcăneanu, Irina Tudose, Beatrice Bălăceanu

**Affiliations:** 1Dermatology Department, “Carol Davila” University of Medicine and Pharmacy, 020021 Bucharest, Romania; 2Dermatology Department, “Elias” University Emergency Hospital, 011461 Bucharest, Romania; 3Orthopedy and Traumatology Clinic, Clinical Emergency Hospital, 014451 Bucharest, Romania; 4Anatomic Pathology Laboratory, “Elias” University Emergency Hospital, 011461 Bucharest, Romania

**Keywords:** dermatofibroma, atypical fibrous histiocytoma, dermoscopy, benign cutaneous tumours

## Abstract

Dermatofibroma (DF) or fibrous histiocytoma is one of the most frequent benign cutaneous soft-tissue lesions, characterized by a post-inflammatory tissue reaction associated with fibrosis of the dermis. Clinically DFs have a polymorphous clinical aspect from the solitary, firm, single nodules to multiple papules with a relatively smooth surface. However, multiple atypical clinicopathological variants of DFs have been reported and, therefore, clinical recognition may become challenging, leading to a more burdensome identification and sometimes to misdiagnosis. Dermoscopy is considered an important tool in DFs diagnosis, as it improves diagnostic accuracy for clinically amelanotic nodules. Although typical dermoscopic patterns are most frequently seen in clinical practice, there have also been some atypical variants described, mimicking some underlying recurrent and sometimes harmful skin afflictions. Usually, no treatment is required, although an appropriate work-up may be necessary in specific cases, such as in the presence of atypical variants or a history of recent changes. This narrative review’s aim is to summarize current evidence regarding clinical presentation, positive and differential diagnosis of atypical dermatofibromas and also to raise awareness about the importance of specific characteristics of atypical variants to better differentiate them from malignant conditions.

## 1. Introduction

Dermatofibroma (DF), also known as fibrous histiocytoma, is a relatively common benign cutaneous tumour characterized by a post-inflammatory tissue reaction associated with fibrosis of the dermis [1,2,3,4,5]. It mostly occurs in young or middle-aged (20 to 40 years old) adults, generally in female patients, although there are histologic variants frequently encountered in males [1,3,6]. DFs with classical morphology have also been described in children aged less than 5 years old [7]. Although various locations have been noticed (head, face, auricle, neck, trunk, shoulder, pelvic girdles, and digits), DFs usually appear on the lower extremities [1,6,7,8,9,10,11,12,13]. DFs are generally asymptomatic but sometimes can become pruritic and tender [4,5]. On palpation, upon lateral compression of the skin, DFs characteristically sink below the level of the skin, a feature also known as the dimple sign [4,5,8,9,14].

The pathogenesis of DFs is unknown, although they usually arise as consequence of local trauma (tuberculin skin testing, skin tattooing, traumatism caused by razor, thorns or wood splinters etc), insect bites or an underlying condition (folliculitis) [2,4,15,16,17,18]. Even though local recurrence and rarely distant metastases have been mentioned in the scientific literature, DFs are considered benign lesions [19]. 

Yamamoto et al. Addressed the role of mast cells in the development of DFs, as they were found in solitary and multiple variants [20]. Mast cells could induce histopathologic changes, such as basal melanosis, acanthosis of the epidermis, and mononuclear cell recruitment [20]. 

Immunohistochemical testing identified the presence of factor XIIIa, which marks dermal dendritic cells [21,22,23,24]. MAC 387, which was labeled histiocytes, did not show relevant results, and the presence of CD68-positive histiocytes was not consistent [21,22,23,24]. One study analyzing 28 cases of dermatofibromas, showed that the majority of spindle-shaped cells, independently of the histological variant, stained positively for HSP47, a marker for skin fibroblasts [21]. Transforming growth factor-beta may also stimulate the fibrosis found in dermatofibromas [25,26].

Other studies suggested that the cell surface proteoglycan, fibroblast growth factor receptor 2, which plays a role in the epithelial–mesenchymal cross-talk, and syndecan-1, may also be involved in the pathogenesis of dermatofibromas [27,28]. Furthermore, CD14+ monocytes have been proposed as the original cells of dermatofibromas [22].

Regarding gene fusion, ALKgene rearrangement and overexpression has been found in both epithelioid and atypical dermatofibromas [29,30,31,32]. As such there have been reported rare autosomal dominant familial cases [2,4].

Reactive tissue alterations and neoplastic proliferation clinical clonality have been suggested as mechanisms involved in the pathogenesis of DFs [33,34]. Spontaneous development, lack of regression and the presence of clonal markers during the analysis of X-chromosome inactivation, may also support the clonal or neoplastic mechanism [19]. Mentzel et al. Investigated 7 cases of clinically aggressive dermatofibromas and underlined the malignant transformation of a cellular dermatofibroma into a spindle cell sarcoma [35]. Chromosomal aberrations by array-comparative genomic hybridization have been proposed as possible diagnostic tools for potentially metastatic dermatofibromas [36].

DFs usually have an excellent prognosis and do not require treatment unless the lesion is changing, bleeding, becomes symptomatic or suspicious, another diagnosis is more probable or the patient demands it clinical cosmetic reasons [4,19]. Complete surgical excision with clear margins for histopathologic examination is the most common therapy [4,19,37]. Atypical variants are more prone to recur and as a result, re-excision might be necessary [37]. Another alternative is liquid nitrogen cryotherapy [37].

## 2. Materials and Methods

A systematic literature search was done in the PubMed, Web of Science Core Collection, and Google Scholar databases, using the terms “atypical dermatofibroma”, and “atypical fibrous histiocytoma”. A total number of 1092 articles, 135 reviews, and 571 case reports were found. All the articles, reviews, and case reports included in the study were limited to English full text in humans. Finally, 134 studies were included in the review. The pictures are from the patients admitted in “Elias” University Emergency Hospital in the period 2021–2022. Written informed consent was obtained from all subjects involved in the study (Figure 1).

## 3. Results

This narrative review aimed to reevaluate the clinical and land dermoscopic patterns of atypical dermatofibromas compared to the typical ones. Although the clinical diagnosis of DFs may be simple in daily practice, in the presence of various patterns, diagnosis of DFs can become challenging. Therefore, specific characterization of these atypical variants is essential in differentiating them from malignant conditions and assessing the risk of local recurrence.

### 3.1. Clinical Presentation

Clinically, DF usually presents as solitary, hyperkeratotic, small (0.3 to 1 cm) and slow-growing nodule with a red-brown surface [2,4,5,19] (Figure 2). The rate of recurrence seems to be higher in lesions initially greater than 1 cm [6]. Other clinical patterns include firm, flat, sometimes atrophic, single or multiple papules, plaques, with a variety of colors (light brown, dark brown, purple, red or yellow) [1,39] (Figure 3).

Colour may also vary depending on the Fitzpatrick fototype (Figure 4). The overlying skin can be pink, red, purple, gray, yellow, orange, blue, brown or black [40]. On palpation DFs have the consistency of a nodule, that moves freely over the subcutis [40]. The dimple sign is valuable in the diagnosis of DFs, although it may not always assure it [41].

Besides the classical clinical presentation, there have also been described some unusual atypical variants. Rare variants may include *metastasizing benign DFs*, which are usually larger than typical variants (more than 3 cm) [4,42,43]. Morphological features can be those of cellular, aneurysmal or atypical DFs and a greater number of mitosis has been noticed in this cases [4,42,43]. Extension into the subcutaneous layer and local recurrence has also been described [4,42,43]. Regarding metastatic sites, lymph nodes and lungs are the most frequent ones [4,42,43].

*Giant lesions* (larger than 5 cm) have also been described in the scientific literature [4,40,43] (Figure 5). The largest tumor reported measured 17 × 9 × 4 cm [44].

*Multiple clustered DFs* (more than 15) appear like a plaque with various single hyperpigmented papules [45,46]. They may also occur in children and can be either congenital or eruptive [45,46]. Atypical fibroxanthoma and dermatofibrosarcoma protuberans are differential dignosis that should be taken into consideration [45,46]. 

*Multiple eruptive, diffuse, and persistent DFs* appear in less than 1% of cases, the majority of patients suffering from an underlying affliction, such as human immunodeficiency virus infection, autoimmune diseases (systemic lupus erythematosus, dermatomyositis, myasthenia gravis, pemphigus vulgaris), Graves disease, Hashimoto thyroiditis, chromosomal alterations (Down syndrome), hematologic malignancies (leukemia, cutaneous T-cell lymphoma, myelodysplastic syndrome, multiple myeloma) atopic dermatitis, metabolic disorders (hypercholesterolemia), glycosuria, hydronephrosis, diabetes mellitus, breast cancer, ulcerative colitis, Crohn’s disease and sarcoidosis [47,48,49,50,51,52,53,54,55,56,57,58,59,60,61]. Moreover, multiple eruptive DFs have been linked to antiretroviral therapy (efalizumab and brentuximab vedotin), tyrosine kinase inhibitors (imatinib), and antitumor necrosis factor-alpha agents [62,63,64,65]. Some cases have also been described in pregnant women [4,66]. 

Other atypical presentations may include *polypoid*, *atrophic*, and DF with spreading *satelitosis* [67,68,69]. 

A Meyerson phenomenon adjacent to the DF has been seen [70] (Figure 6). 

### 3.2. Diagnosis and Assessment

As multiple atypical variants of DF have been reported in the literature, clinical recognition may become challenging, leading to a more burdensome identification and sometimes to misdiagnosis [2,9]. Although diagnosis is commonly based on clinical presentation and history, further diagnostic tools such as dermoscopy, variable–frequency ultrasonography, fluorodeoxyglucose positron–emission tomography (FDG-PET) scans, and confocal laser scanning microscopy are necessary.

**Dermoscopy** is a non-invasive procedure useful for the diagnosis and management of pigmented tumours of the skin [9,71]. Among the scientific literature, various dermoscopic structures have been mentioned. DFs typical dermoscopic appearance includes the presence of a delicate, peripheral light-to-medium brown pigment network with a sharply demarcated central white scar-like area, white network and homogeneous pigmentation [1,8,9] (Figure 7). 

For atypical variants, high definition optical coherence tomography can be useful as it correlates with histopathological types of DFs [72]. As DFs proved to have intense F-18 fluorodeoxyglucose uptake on positron emission tomography-computed tomography scan, the letter coud be a possible diagnostic alternative [73]. Although confocal laser scanning microsocopy is mainly used to evaluate melanocytic lesions, it is helpful in diagnosing DFs as their features (bright rings at the periphery, collagen structures at the center, central keratin) correlate with both dermoscopy and histopathology [74]. 

Although typical dermoscopic patterns are most frequently seen in clinical practice, some authors described atypical types of DFs (Table 1).

*Additional clinical features* may include ring-like or donut-shaped globular structures, vascular structures and sometimes ulceration, comedo-like openings, scale, crusts, or peripheral collarette fissures [8,9].

The *pigment network* may vary from peripheral/total/irregular delicate to peripheral/total/irregular prominent and atypical appearance [1,3,9]. Aytekin et al. evaluated dermoscopically 142 DFs of 72 patients and concluded that pigment network was found in 57% of cases, the most common subtype being irregular delicate or asymmetric pigment network [1]. Delicate pigment network seen in DFs is commonly thin, varying from light to medium brown and it is considered that it results as hyperpigmentation of rete ridges rather than the proliferation melanocytes [1]. According to the study performed by Arpaia et al., the pigment network was darker in the center, becoming gradually pale towards the periphery with brownish thin streaks [80]. In 2000, Ferrari et al. noticed that the peripheral pigment network and central white scar-like patches are more prevalent in women and among the classical histopathologic type of DFs [36]. The results may alternate due to the quality of the dermoscopy (contact/non-contact, polarized/nonpolarized light) and the difference among genders [1]. Interestingly, Zaballos et al. also identified in some DFs a significant and/or atypical pigment network [8].

*Central white scar-like patches* are sharply demarcated with irregular white regions, histopathologically characterized by various grades of fibrosis in the dermis [1,9] (Figure 8).

Aytekin et al. have detected white scar-like patches in 37.3% of cases, the most common subtype being the central ones [1]. This structure is considered the most widespread feature of diffuse fibrous DF with a peripheral delicate pigment network [1]. It has also been noticed that the scar structure occasionally got a white radial streaks appearance, which gave the aspect of a spitzoid pattern [1]. Zaballos et al. evaluated 412 DFs and observed that white scar-like patches are mainly localized in the center part of the lesion [8]. Moreover, Arpaia et al. concluded that the central white patch was the most frequent dermoscopic feature, observed in 91.6% of cases [80].

The *white network* may be central, total, irregular, or crystalline-like [1,3,9]. Zaballos et al. identified a network of white lines and brown holes, which was later considered a variation of the white scar-like patch [8]. There has been raised awareness of the importance of distinguishing this structure from dysplastic nevi, Spitz nevi and the negative pigment network encountered in melanomas [1,8].

The *homogenous pigmentation* may include multiple colors (brown, yellow) or it may appear as hypopigmentation [1,3,9] (Figure 9).

Ferrari et al. noticed that homogeneous pigmentation was most frequently in females and DFs with sebaceous hyperplasia, whereas peripheral homogeneous pigmentation was mostly encountered in men [36]. Karaarslan et al. observed a homogeneous blueish pigmentation that was associated with the hemosiderotic type of the DF [81]. Usually, hemosiderotic variants are indicated by the green color [79].

The *vascular pattern* has been widely discussed. Vascular structures are used in dermoscopy to diagnose melanoma and other pigmented or vascular tumors, which mimic melanoma [1,3]. Nevertheless, DFs may have peripheral, central, or total erythema, dotted, hairpin, glomerular, comma, or linear vessels, but also polymorphic and atypical ones [1,3,9] (Figure 10).

Contrary to other studies, Genc et al. found vascular structures to be the most frequent dermatoscopic feature and described a red to brown halo phenomenon in 4.9% of DFs [75]. Another study performed by Agero et al. concluded that blood vessels were seen in 44% of DFs when using polarized light [82]. Ferrari et al. described 2 DFs with dotted vessel patterns, whereas Aytekin et al. stated that the most frequent vascular structures in their study were erythema and dotted vessels [1,9,36].

Other not so common dermatoscopic changes may involve ring-like structures, ulceration, scales, fissures, milia-like cysts, hemorrhage, crusts or white radial streaks [1,3,9,82]. Genc et al. have conducted a study which classified DFs depending on the dermatoscopic similarities to other lesions [75]:Melanoma-like: various colors and patterns, white structureless areas, polarizing-specific white lines, pink-red or blue-gray structureless areas, dark brown thick reticular lines, peripheral black clods and eccentric distribution of straight, curved, dotted and branched vessels [75].Basal cell carcinoma-like: arterial structures specifically in the papillary dermis mostly at the periphery of the lesion [75] (Figure 11).Keratoacanthoma-like: central keratin area with a surrounding radial arrangement of polymorphic vessels (curved, branched and dotted) [75].Seborrheic keratosis-like: thick curved lines, orange, brown or white clods, brown-black crusted structures, blue-gray structureless areas and loop, dotted or coiled vessels [75].Nevus-like: various hypopigmented structureless areas, having in between multifocal thin brown reticular lines [75] (Figure 4 and Figure 12).Nevus sebaceous-like: white lines (associated with dermal fibrosis), peripheral thin brown reticular lines and central large yellow clods (associated with sebaceous hyperplasia) [75] (Figure 13).Xanthogranuloma-like: yellow structureless areas, coiled vessels with a peripheral reddish halo [75].Pyogenic granuloma-like: polymorphic vessels (curved, dotted, straight and branched) with irregular distribution, white lines and pink-red structureless areas [75].Spitzoid-like: pink-red structureless areas, shiny white lines, white structureless areas, light brown clods, halo phenomenon, dotted vessels [75].

Llambrich et al. performed a retrospective review, analyzing clinical and dermoscopic features of 36 pink nodular DFs [3]. They underlined the importance of a correct differential diagnosis as pink nodular lesions with erythema, vascular structures, shiny white streaks and a central white patch may suggest malignancy, mainly amelanotic/hypomelanotic melanoma [3]. Moreover, regarding the dermopathological types, non-fibrocollagenous variants of DFs were proned to have atypical patterns [36,75].

Melanoma-like and pyogenic granuloma-like atypical patterns were seen mostly in the case of aneurysmal DFs [75]. Furthermore DFs may have a pinkish-red pigmentation, dotted vessels and superficial white scales resembling psoriasis [9]. A “collision tumour-like” pattern was also described having a white area with focal pigment network [9]. Particularly, collision-like patterns along with melanoma-like and vascular tumour-like patterns were most commonly noticed in men [9]. Aditionally there have been described palisading, granular cell, myxoid, lichenoid, balloon cell and signet-ring cell variants [2,6,9].

As such, dermoscopy may be beneficial in increasing diagnosis and management accuracy, but since dermoscopic features may vary as well, it is certainly important to take into consideration other differential diagnoses [4,9,71,75].

### 3.3. Histologic Variants

Histologically, DFs contain uniform spindle cells organized in elongated fascicles [4]. Classical histopathological features of typical DFs include an overlying achantotic, hyperkeratotic and sometimes hyperpigmented epidermis [19,83,84,85,86,87,88] (Figure 14). 

Moreover, the epidermis usually exhibits elongated rete ridges containing hyperpigmented basal keratinocytes, aspect known as “dirty feet” sign [2]. There is also a proliferation of spindle-shaped fibrous cells mixed with histiocytoid cells at the level of the dermis [22,89,90]. Collagen bundles are commonly seen between the spindled fibrous cells along with the unaffected layeser, known as the “Grenz zone” [2,89,90,91].

The scientific literature has classified various types of DFs, regarding their histopathological characteristics [2]. Histological features may coexist in the same lesion [87]. As such there have been mentioned a lot of histopathological types: fibrocollagenous, cellular, keloidal, atrophic, aneurysmal, storiform, fibrocollagenous with sebaceous induction, lipidized, hemosiderotic, epithelioid, lichenoid, baloon cell, signet-ring myofibroblastic, clear cell, palisading, granular cell, myxoid and also the atypical type [2,6,71,83,84,85,87,88].

A histologic review performed by Alves et al. on 192 dermatofibromas stated that common fibrous hystiocitoma was the most frequent type, observed in 80% of cases [2]. Individual collagen bundles encompassed by lesional cells (fibroblasts, macrophages and blood vessels) and a predominantly lymphocytic inflammatory infiltrate may be seen [2].

Atypical DFs also known as DFs with monster cells, are poorly documented variants of typical fibrous histiocytomas [88]. Besides typical findings, atypical DF are comprised of pleomorphic spindle-like, hystiocite-like cells and multinucleate giant cells. [89].

LeBoit and Barr firstly described dermatofibroma with granular cells in 1991 [87,88]. This rare histologic variant can be confused with other malignant or benign cutaneous neoplasms such as: benign granular cell tumor, malignant granular cell tumor, primitive polypoid granular cell tumor, granular cell ameloblastoma, granular cell fibrous papule of the nose, granular cell basal cell carcinoma, granular cell schwannoma, granular cell leiomyoma, granular cell leiomyosarcoma or angiosarcoma and granular cell dermatofibrosarcoma protuberans [87]. Morfology of the lesion along with immunohistochemical evaluation might sometimes be decisive for the corect diagnosis [86]. For instance, benign granular cell tumors are positive for S-100 protein, CD63, CD68 and neuron-specific enolase whereas atypical fibroxanthoma stains negatively for S100 protein, Melan-A, human melanoma black (HMB)-45 pan-cytokeratin (CK) and actin and positively for CD68 and vimentin [87].

Immunohistochemistry can be useful to differentiate DFs from schwannomas, leiomyomas and leiomyosarcomas [87].

### 3.4. Differential Diagnosis

It is extremely important to recognize atypical DFs, as *cutaneous melanoma* is a vital clinical differential diagnosis and may display similar characteristics [9,19]. Nevertheless, other afflictions as well may be taken into accounts, such as intradermal nevi, basal cell carcinomas, keratoacanthomas, and dermatofibrosarcomas protuberans [4]. Differential diagnoses may also include angiokeratomas, Spitz-nevi, melanocytic nevi, blue nevi, granuloma annulare, supernumerary nipple, acrochordon, atypical fibroxanthoma, cutaneous metastasis, cutaneous T-cell lymphoma, cylindroma, pilomatrixoma or targetoid hemosiderotic hemangiomas [14,92,93].

*Dermatofibrosarcoma protuberans (DFSP)* appears as painless, slow-growing skin-colored nodule, with a finger-like projections pattern and should be distinguished from benign DFs as it is locally aggressive [4,19,94]. A delayed accurate diagnosis leads to clinical pitfalls [94]. A more cellular appearance and a “honeycomb” display of the subcutaneous fat is often seen in DFSP [19]. Immunohistochemical staining is also very useful, as there are various markers to differentiate the two **entities** [19,95,96,97,98,99,100,101,102,103]. Although DFSP stains positive for CD-34, nestin and collagen triple helix repeat containing-1 (Cthrc1) and negative for factor XIIIa, there has also been noticed an elevated expression of thrombospondin-1 (TSP-1) [19,98,100,101,103]. DFSP characteristically has a genomic reciprocal translocation in t (17;22) (q22;q13) that causes the fusion of the platelet-derived growth factor B-chain (PDGFB) and the promoter of the collagen type Iα1 (COL1A1) genes and might be detected by fluorescent in situ hybridisation or real-time PCR [94].

In comparison, DF stains positive for factor XIIIa, D2-40, insulin-like growth factor–binding protein 7 (IGFBP7), cathepsin K, CD99, leukocyte-specific protein 1 (LSP1) and 5-hydroxymethylcytosine (5-hmC) and negative for CD-34 [19,26,37,99,102,104,105,106,107]. Occasionally, the cellular type of DF may stain positive for CD34 [95,96]. Stromelysin-3 (ST-3) expression of DF shall also help to differentiate it from DFSP [97]. FGFR3/FOXN1 and FGF2/FGFR4 expression in the pathogenesis of DF is practical [104]. Fluorescence in situ hybridization (FISH) analysis is a valuable tool as well [108]. B-cell lymphoma 2 (Bcl-2) expression, autophagy marker Atg5, and phosphohistone-H3 can help to differentiate between DF and DFSP [109,110]. Moreover, Ki-67 staining shows a higher proliferation index in the case of DFSP [19].

Hemosiderotic dermatofibromas, dermatoscopically characterized by a blue/red center with white lines and maybe network and vessels at the periphery may lead to a dermoscopically differential diagnosis with *Kaposi sarcoma* due to the intense vascularity [28,29]. Nevertheless, Kaposi sarcoma stains are positive for CD31, CD34 and D2-40, and patients are also positive for HHV-8 [19].

CK20 positive Merkel cells, present in the follicular induction, crowding, no peripheral palisading, clear cell hyperplasia, and the absence of nuclear atypia are helpful pathologic features in differentiating DFs with follicular induction from *basal cell carcinomas* [19].

## 4. Discussion

This narrative review aimed to reevaluate the clinical and dermoscopic patterns of atypical dermatofibromas compared to the typical ones. Moreover, there have been mentioned some not-so-common etiopathogenic factors. Dermatofibromas are prevalent cutaneous benign tumours that most frequently affect young or middle-aged adults. Clinically, dermatofibromas appear as single or multiple firm papules or nodules with a smooth surface anywhere on the body, mainly on the lower extremities. They can vary in size and colour from light brown to dark brown, yellow, purple or red. Although the clinical diagnosis of DFs may be simple in daily practice, in the presence of various patterns, diagnosis of DFs can become challenging.

Therefore, it is essential to consider the possible links between dermoscopy and histology and complete surgical excision, especially in the presence of atypical variants or a history of recent changes. Thus, the precise definition of dermoscopic patterns for this frequent benign tumour is of major interest.

In a reverse manner, the possibility of a misdiagnosis of malignant skin disorders, inlcuding non-melanoma skin cancer, is a main challenge in terms of worldwide public health management. A lot of clinicians may face it and thus malignancy-related misdiagnosis remains one of the main issues in the dermatologic field. Similar to the case of melanomas or basal cell carcinomas being misdiagnosed as diabetic foot ulcer, they can be more easily considered, even by experts, as benign lesion, such as dermatofibromas [111]. As malignant tumors may sometimes mimic benign conditions, the main focus has been on finding a non-invasive, reliable, sensible and highly specific diagnostic methods to identify specific features which suggest malignancy.

### Promising Differential Diagnostic Methods

Apart from dermoscopy and digital dermoscopy, not so common diagnostic tools may include variable–frequency ultrasonography, in which DFs appear as hypoechoic solid nodules and high-definition optical coherent tomography, in which DFs can resemble malignant conditions on FDG-PET scans [73,74]. Among other studied techniques there have been mentioned reflectance confocal microscopy (RCM), multiphoton microscopy, fluorescence evaluation, Raman spectroscopy and diffuse reflectance [112,113,114,115]. High-frequency ultrasonography has been a valuable diagnostic and prognostic tool in early detecting other types of malignancies, such as hepatocellular carcinoma, for a long time, but it recently has begun to represent a promising opportunity in dermatology by using deep learning-based algorithms to analyse automated images [77,116,117]. Complementary techniques, such as dermoscopy in conjunction with RCM, may also enhance diagnostic accuracy of melanocytic conditions [118,119,120]. Confocal laser scanning microscopy is also considered an alternative tool [118,119,120].

The use of fractal parameters and fractal analysis method in dermatology is promising in the evaluation of image parameters, independently of the adopted scale [121]. There are currently studies being carried regarding the usefulness of fractal parameters in building classes of disease units based upon pictures of cutaneous pigmented lesions [121,122]. It might provide fully automatic diagnostic systems able to determine the type of pigmented tumor and inform us regarding the most adequate management [121].

Taking into consideration that dermatology is a largely visual speciality, the high cost of travel expenses to urban centres, the long wait times to see a dermatologist and the shortage of dermatology services mainly in rural areas, several research studies analizying the role of an artificial intelligence (AI) system as a diagnostic tool for the management of skin conditions have been conducted, focusing mainly on the malignant ones [123,124,125,126,127,128,129]. Several studies proved that telehealth platforms, easily reachable through smartphone apps, could increase patients access to dermatological care, especially during COVID-19 pandemic [127,130,131,132]. Some of the apps can use AI to provide various differential diagnoses depending on the information provided: patient demographics, lesion type, location, symptoms and progression [127,129]. Artificial neural networks, such as convolutional neural networks (CNNs), can be used to analyse visual imagery, being very effective in recognizing automated images and equal or superior than dermatologist in recognizing skin cancers [123]. Implementing AI as a diagnostic aid in the clinical practice may be safe, useful and feasible for skin lesions accurate detection and for better differentiating malignant from benign ones [123,124,125,126,127].

In order to diminish the high degree of subjectivity and variability regarding specificity, sensitivity, and diagnostic accuracy when performing RCM, a lot of artificial intelligence algorithms were created to ensure alternatives, assistance and support od dermatologists on a daily basis [118]. AI in RCM has been used so far to point out the dermal–epidermal junction, evaluate the the quality of RCM mosaics and distinguish between different skin tumors [118].

A lot of other studies have analyzed the correlation between spectrophotometric parameters of skin color and behavioral/environmental factors to predict the risk of cutaneous malignancies [133]. They concluded that the measurment of skin melanin index measured on the arm or buttock is the simplest predictor and should be added in predictive models. Regarding the environmental/behavioral factors, the total number of sunburns appear to be the most important one. As such, spectrophotometric measurements may be considered a quick screening examination method of the skin [133].

In histopathology, AI is efficient in classifying and characterizing tissues, in detecting mitosis and segment histologic primitives as epithelium, nuclei and tubules [128].

As a consequence of complexity and intransparency of deep neural networks in classifying skin cancer, explainable artificial intelligence (XAI) has also been suggested as an alternative although further research studies are needed to evaluate the influence of XAI in detecting cutaneous cancer [134].

There are lot of opportunities that lie ahead, from automated classification of cutaneous cancer through convolutional neural networks, sequential digital dermoscopy and automated total body photography to AI and automated teledermoscopy [135]. However, the potential use of AI in clinical practice remains to be addressed due to their limitations and further studies need to be conducted in order to implement it every day medical practice [135].

## 5. Conclusions

Clinical diagnosis of typical dermatofibromas is easy, with a classic dermoscopic pattern of pigmented network and central white patch. However, in current clinical situations, dermatofibromas display a wide range of presentations and histological variants that make the differentiation from other tumours, such as malignant melanoma, very difficult. Specific characterization of these atypical variants is essential in differentiating them from possibly more aggressive lesions and assessing the risk of local recurrence. As a matter of fact, the definitive diagnosis of a skin condition, especially in a doubtful clinical diagnostic scenario, demands complete surgical resection and histopathological analysis. As artificial intelligence technologies had reached an impressive precision in identifying various skin lesions, along with other inovative diagnostic methods, we can emphasise that in the future it will lead to improved safety and patient care and maybe enhance dermatologists’ productivity.

## Figures and Tables

**Figure 1 diagnostics-13-00671-f001:**
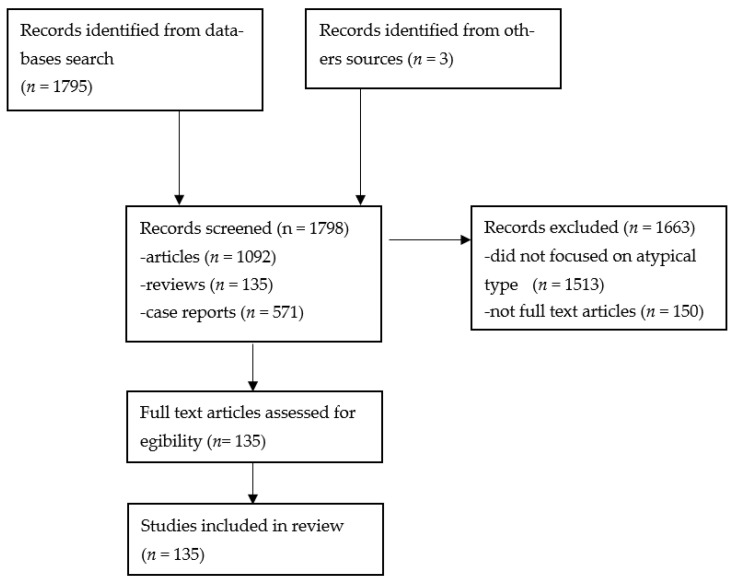
PRISMA flowchart [38].

**Figure 2 diagnostics-13-00671-f002:**
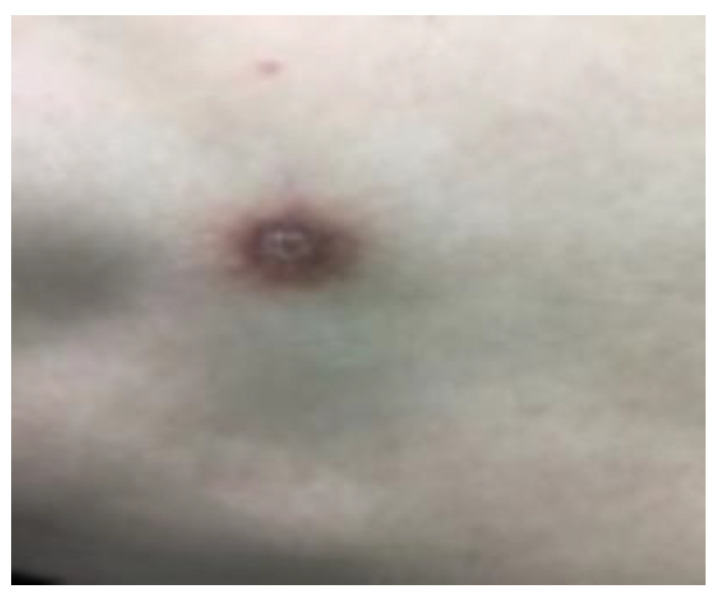
Clinical appearance of a DF in a young woman: a solitary, well-defined, hyperkeratotic nodule with a diametre of about 1 cm with a yellow-brown surface.

**Figure 3 diagnostics-13-00671-f003:**
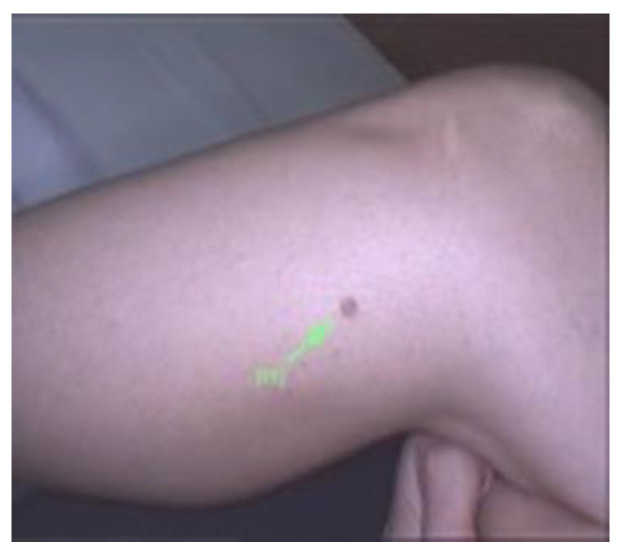
Another clinical pattern of a DF located on the leg: a flat, light brown, single papule.

**Figure 4 diagnostics-13-00671-f004:**
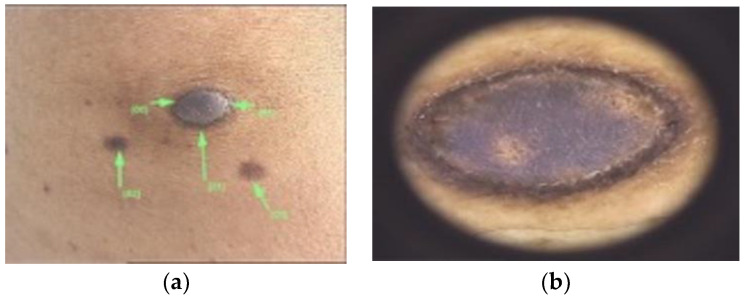
(**a**) DF with a nodular, blue appearance in a 4th Fitzpatrick phototype patient. Differential diagnoses may include a blue nevus. (**b**) Dermoscopic image of a nodular, blue DF, with well-defined borders and some scales.

**Figure 5 diagnostics-13-00671-f005:**
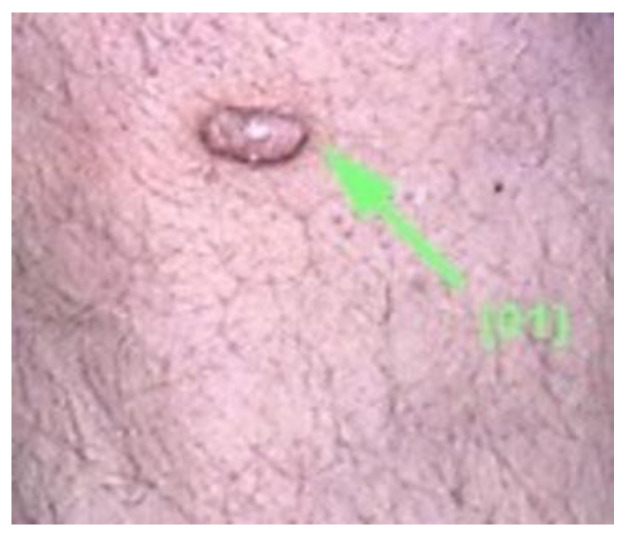
Giant DF in a young patient, with a diameter of about 5.5 cm.

**Figure 6 diagnostics-13-00671-f006:**
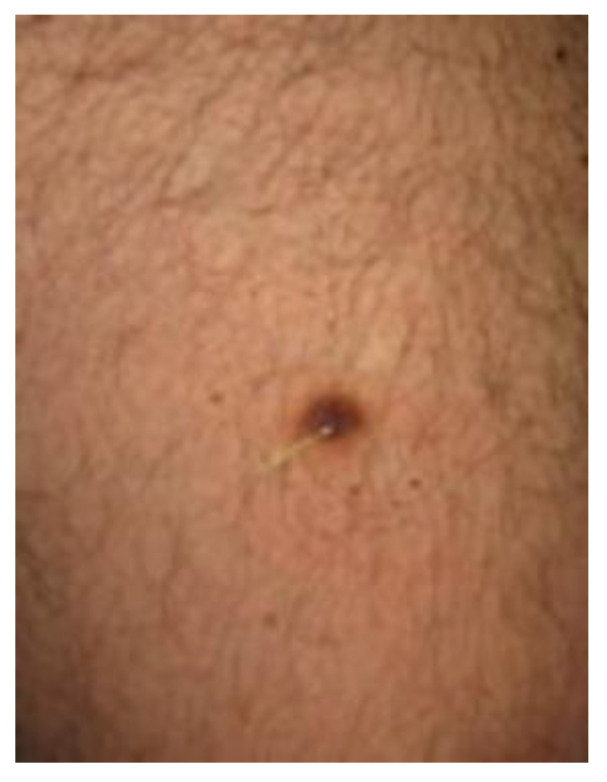
A Meyerson phenomenon is adjacent to the DF.

**Figure 7 diagnostics-13-00671-f007:**
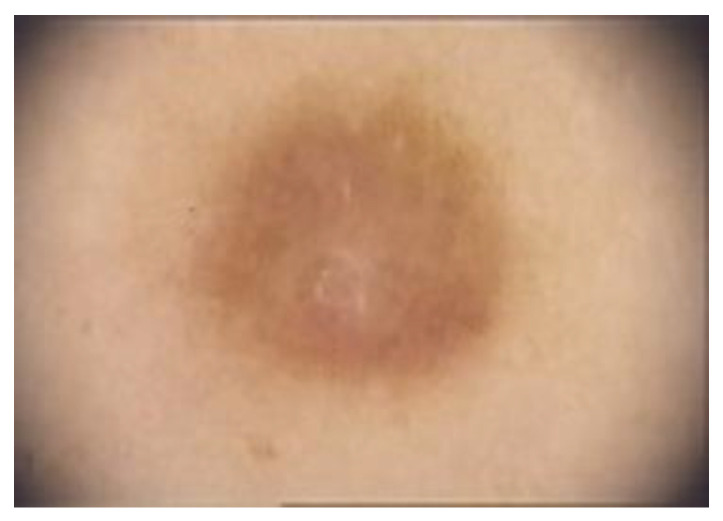
Typical dermoscopic appearance of a DF: delicate, peripheral light-to-medium brown pigment network with a sharply demarcated central white scar-like area, white network and homogeneous pigmentation.

**Figure 8 diagnostics-13-00671-f008:**
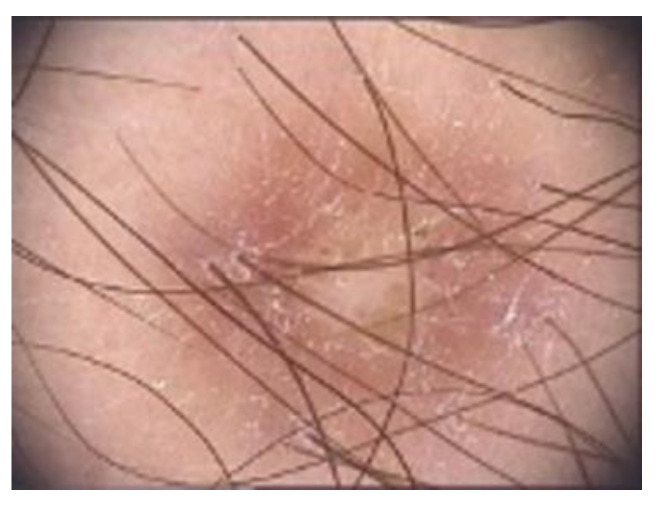
The dermoscopic appearance of central white scar-like patches sharply demarcated with irregular white regions.

**Figure 9 diagnostics-13-00671-f009:**
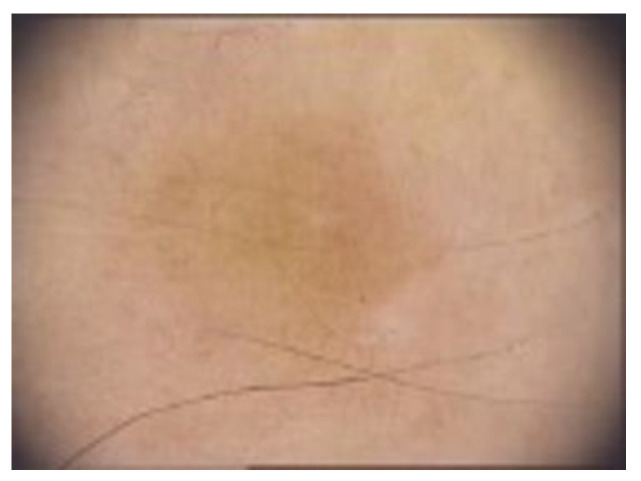
The dermoscopic appearance of a homogenous yellow-brown pigmentation of a DF.

**Figure 10 diagnostics-13-00671-f010:**
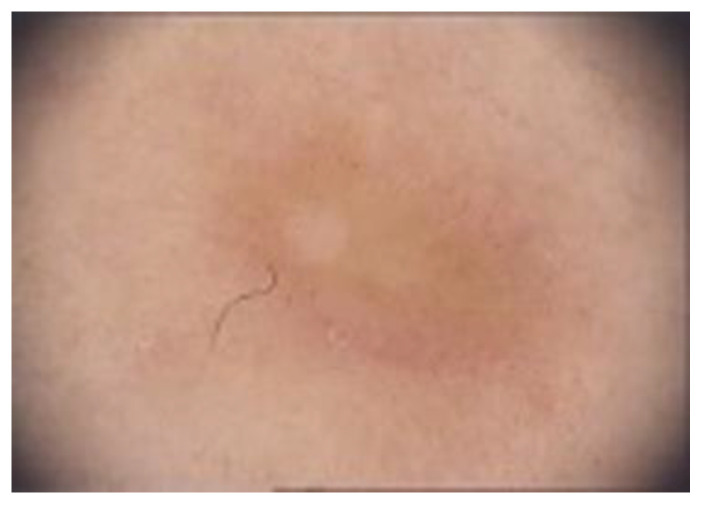
The dermoscopic appearance of a DF with peripheral erythema and dotted vessels.

**Figure 11 diagnostics-13-00671-f011:**
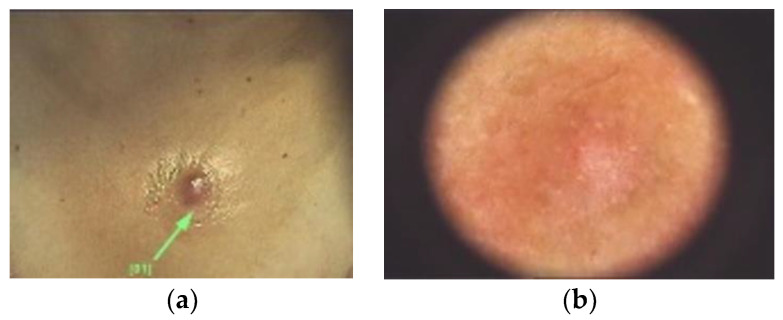
(**a**) Clinical image of a DF with a solitary, well-defined, nodular, pink appearance in a female patient. Differential diagnoses may include basal cell carcinoma. (**b**) Dermoscopic image of a DF with arborizing vessels, along with the central white scar-like patch and fine delicate pigment network.

**Figure 12 diagnostics-13-00671-f012:**
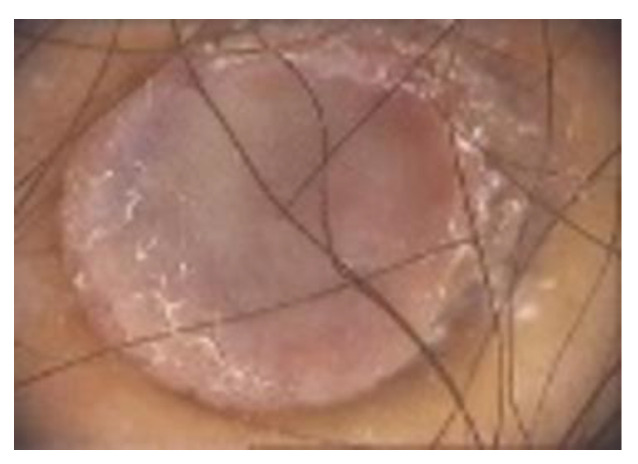
Dermoscopic image of a nevus-like DF, but also with coiled vessels and some scales.

**Figure 13 diagnostics-13-00671-f013:**
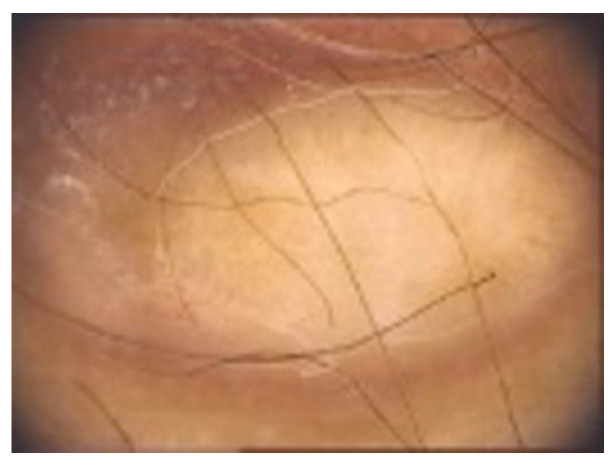
Dermoscopic image of a DF with a nevus sebaceous-like appearance: white lines (associated with dermal fibrosis), peripheral thin brown reticular lines and central yellow structures.

**Figure 14 diagnostics-13-00671-f014:**
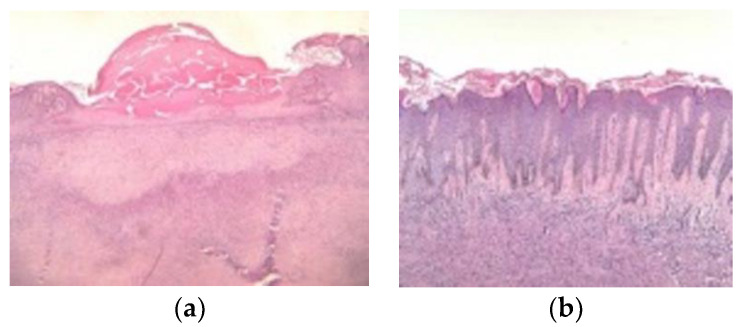
(**a**) Histopathologic examination (hematoxylin and eosin, ×10): tumour proliferation localized in the papillary dermis and extending to the deep dermis, with interspersed collagen bundles, separated from the epidermis by a grenz zone. The overlying epidermis presents erosions centrally and collections in the keratin layer. (**b**) Histopathologic examination (hematoxylin and eosin, ×10): tumour proliferation composed of elongated and spindle-shape cells with elongated nuclei, in a fascicular-storiform configuration localized in the papillary dermis and extending to the deep dermis. The overlying epidermis has a hyperplastic appearance with hyperorthokeratosis, acanthosis, and elongation of the rate ridges. There is also follicular induction at the epidermis level.

**Table 1 diagnostics-13-00671-t001:** Dermoscopic features of atypical dermatofibromas.

Study, Authors	Article Type	Year of Publication	Atypical Features with High Risk	Cases (*n*/%)
Aytekin S et al. [1]	Original article	2021	Irregular delicate/asymmetric pigment network Peripheral proeminent pigment network Irregular proeminent pigment network Atypical pigment network Irregular white network Irregular brown areas Dotted vessels Glomerular vessels Polymorphous/atypical vessels Ulceration White radial streaks	31 (21.8) 3 (2.1) 2 (1.4) 2 (1.4) 3 (2.1) 9 (6.3) 34 (23.9) 2 (1.4) 6 (4.2) 5 (3.5) 8 (5.6)
Genc Y et al. [75]	Report	2020	Melanoma-like BCC-like Keratoachantoma-like Spitzoid-like	11 (19.4) 3 (4.9) 2 (2.6) 26 (19.4)
Llambrich A et al. [3]	Research letter	2019	Dotted vessels Arborizing vessels Polymorphous/atypical vessels Shiny white streaks	18 (50) 7 (19.4) 18 (50) 16 (44.4)
Lin MJ et al. [76]	Original research	2018	Dotted/pinpoint vessels Sharply focused arborizing vessels Linear irregular vessels Glomerular vessels Polymorphous vascular pattern White lines Ulceration Blue/grey veil	2 (22) 0 1 (11) 1 (11) 1 (11) 0 0 0
Won KY et al. [77]	Original research	2017	Irregular shape Spiculated margins	8 (44) 12 (67)
Kelati A et al. [78]	Research article	2017	White streaks Ulceration Brown streks Negative-network-like appearance Dotted vessels Multicomponent melanoma-like Vascular tumor-like BCC-like Collision tumor-like Peripheral diffuse pink to red to reddish violet halo White ring around an ulceration Pink bluish pigmentation with vascularization Pigment network with a ring around follicular opening	18 (18) 6 (6) 6 (6) 3 (3) 23.3% 20 (20%) 0 0 0 7 (7%) 6 (6%) 7 (7%) 2 (2%)
Marinescu SA et al. [71]	Case report	2016	Pinky-milk areas Peripheral pigment network Polymorphous atypical vessels	
Roldán-Marín R et al. [79]	Case report	2014	Grey-green colour	
Ferrari A et al. [9]	Original article	2013	Melanoma-like Vascular tumour-like BCC-like Collision tumour-like	21 (16.2) 6 (4.6) 5 (3.8) 3 (2.3)
Zaballos et al. [8]	Prospective study	2008	Proeminent atypical pigment network Irregular pigment network Irregular white network Irregular brown areas Dotted vessels Glomerular vessels Polymorphous/atypical vessels Ulceration	13 (3.1) 8 (1.9) 3 (0.7) 125 (30.6) 3 (0.7) 10 (2.4) 18 (4.4)

## Data Availability

This review summarizes data reported in the literature and it does not report primary data.

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
