# Peer review of "Challenging Patterns of Atypical Dermatofibromas and Promising Diagnostic Tools for Differential Diagnosis of Malignant Lesions"

_diagnostics, 2023, doi:10.3390/diagnostics13040671_

Round 1

Reviewer 1 Report

The authors described and summarized previous dermatofibromas researches and tried to brief an update of these studies regarding pathophysiology, clinical presentation, diagnosis, and treatment of both typical and atypical dermatofibromas.

- This review is only focused on describing "atypical" dermatofibromas compared to the typical one. The title should be with "atypical" and overall clinical/dermatoscopic patterns should be clearly described using tables. It's quite hard to read through and not well organized.

- What is the conclusion/un update? The authors searched the pubmed and other web-based databases for "atypical dermatofibroma", and "atypical fibrous histiocytoma", for what? there are no case studies summarized from other studies.

- Figures are randomly arranged and the source of the figures are not clear. What does "the authors' own archive" mean? Which hospital/when/why these pictures are taken?  It would be easier to see if these are catalogued (systematically listed) with clinical/dermatoscopic patterns.

Author Response

Dear Reviewer 1,

Thank you for the recommendations. We have modified the title of the article and the text in accordance with your recommendations.

„The authors described and summarized previous dermatofibromas researches and tried to brief an update of these studies regarding pathophysiology, clinical presentation, diagnosis, and treatment of both typical and atypical dermatofibromas.”

-„ This review is only focused on describing "atypical" dermatofibromas compared to the typical one. The title should be with "atypical" and overall clinical/dermatoscopic patterns should be clearly described using tables. It's quite hard to read through and not well organized.”

We changed the title according the recommendations and organized in table the dermatoscopic patterns of the atypical lesions.

- „What is the conclusion/un update? The authors searched the pubmed and other web-based databases for "atypical dermatofibroma", and "atypical fibrous histiocytoma", for what? there are no case studies summarized from other studies.”

However, in current clinical situations, dermatofibromas display a wide range of presentations and histological variants that make the differentiation from other tumours, such as malignant melanoma, very difficult.

Specific characterization of these atypical variants is essential in differentiating them from possibly more aggressive lesions and assessing the risk of local recurrence.

As a matter of fact, the definitive diagnosis of a skin condition, especially in a doubtful clinical diagnostic scenario, demands complete surgical resection and histopathological analysis.

As artificial intelligence technologies had reached an impressive precision in identifying various skin lesions, along with other inovative diagnostic methods, we can emphasise that in the future it will lead to improved safety and patient care and maybe enhance dermatologists’ productivity.

- „Figures are randomly arranged and the source of the figures are not clear. What does "the authors' own archive" mean? Which hospital/when/why these pictures are taken?  It would be easier to see if these are catalogued (systematically listed) with clinical/dermatoscopic patterns.”

Institutional Review Board Statement: The study was approved by the local Ethical Committee of the “Elias” University Emergency Hospital. The research was conducted according to the Helsinki Declaration. The pictures are part of the authors' own archive,  and are from the patients admitted in “Elias” University Emergency Hospital in the last 2 years.  They do not contain personal data.

Informed Consent Statement: Informed consent was obtained from all subjects involved in the study.

The figures were arranged in the text.

Best Regards,

Beatrice Balaceanu, MD

Reviewer 2 Report

1. The presented paper aims to summarize current evidence on pathophysiology, clinical presentation, diagnosis, and treatment of both typical and atypical dermatofibromas and also to raise awareness of the importance of specific characteristics of atypical variants to better differentiate them from malignant conditions. The misdiagnosis of malignant skin disorders is a highly dangerous clinical problem that requires new methods and approaches.

2. The Introduction section is extremely brief and should be expanded.

3. The authors should consider the possibility of a misdiagnosis of malignant skin disorders, including parameters of high-risk patients.

- Lyundup, A.V.; Balyasin, M.V.; Maksimova, N.V.; Kovina, M.V.; Krasheninnikov, M.E.; Dyuzheva, T.G.; Yakovenko, S.A.; Appolonova, S.A.; Schiöth, H.B.; Klabukov I.D. Misdiagnosis of diabetic foot ulcer in patients with undiagnosed skin malignancies. International Wound Journal 2022, 19(4): 871-887. https://doi.org/10.1111/iwj.13688

- Kelati, A., Aqil, N., Baybay, H. et al. Beyond classic dermoscopic patterns of dermatofibromas: a prospective research study. J Med Case Reports 2017, 11, 266. https://doi.org/10.1186/s13256-017-1429-6

4. The Materials and Methods section is also extremely brief. The PRISM diagram should be performed and described.

5. The figures in the document are not organized: I recommend combining all of the pictures into some collages. 

6. The paper should discuss the frontier for promising diagnostic methods based on spectrophotometry, AI, and neuronal networks application.

7. The reference list should be expanded to include sources from the last years (2019-2022).

Author Response

Dear Reviewer 2,

Thank you for the recommendations. We have modified the title of the article and the text in accordance with your recommendations.

  1. The presented paper aims to summarize current evidence on pathophysiology, clinical presentation, diagnosis, and treatment of both typical and atypical dermatofibromas and also to raise awareness of the importance of specific characteristics of atypical variants to better differentiate them from malignant conditions. The misdiagnosis of malignant skin disorders is a highly dangerous clinical problem that requires new methods and approaches.”
  2. „The Introduction section is extremely brief and should be expanded.”

We expanded the Introduction section.

  1. „The authors should consider the possibility of a misdiagnosis of malignant skin disorders, including parameters of high-risk patients.”

- Lyundup, A.V.; Balyasin, M.V.; Maksimova, N.V.; Kovina, M.V.; Krasheninnikov, M.E.; Dyuzheva, T.G.; Yakovenko, S.A.; Appolonova, S.A.; Schiöth, H.B.; Klabukov I.D. Misdiagnosis of diabetic foot ulcer in patients with undiagnosed skin malignancies. International Wound Journal 2022, 19(4): 871-887. https://doi.org/10.1111/iwj.13688

In a reverse manner, the possibility of a misdiagnosis of malignant skin disorders, inlcuding non-melanoma skin cancer, is a main challenge in terms of worldwide public health management. As malignant tumors may sometimes mimic benign conditions, the main focus has been on finding a non-invasive, reliable, sensible and highly specific diagnostic methods to identify specific features which suggest malignancy.

- Kelati, A., Aqil, N., Baybay, H. et al. Beyond classic dermoscopic patterns of dermatofibromas: a prospective research study. J Med Case Reports 2017, 11, 266. https://doi.org/10.1186/s13256-017-1429-6

Reference 80 (Kelati at al.) was inserted in the text and table.

  1. „The Materials and Methods section is also extremely brief. The PRISM diagram should be performed and described.”

The PRISMA diagram was performed and described.

  1. „The figures in the document are not organized: I recommend combining all of the pictures into some collages.”

The figures were arranged in the text.

  1. „The paper should discuss the frontier for promising diagnostic methods based on spectrophotometry, AI, and neuronal networks application.”

In the section „Discussion” we introduced the promising differential methods, as spectrophotometry, AI, and neuronal networks application.

  1. „The reference list should be expanded to include sources from the last years (2019-2022).”

We have completed the list of references with all the articles related to the subject from the period 2019-2022.

Best Regards,

Beatrice Balaceanu, MD

Round 2

Reviewer 1 Report

The manuscript is greatly improved and acceptable for publication.  

Author Response

Dear Reviewer 1,

Thank you for the recommendations and guidance. We have modified the title of the article and the text in accordance with your recommendations.

Best Regards,

Beatrice Balaceanu, MD

Reviewer 2 Report

The authors have responded to most of my comments and made the necessary changes to the manuscript. However, the authors deliberately avoid discussing the problem of malignancy-related misdiagnosis. I do believe this issue is worth revealing the paper.

Author Response

Dear Reviewer 2,

Thank you for the recommendations. We have modified the title of the article and the text in accordance with your recommendations.

„The authors have responded to most of my comments and made the necessary changes to the manuscript. However, the authors deliberately avoid discussing the problem of malignancy-related misdiagnosis. I do believe this issue is worth revealing the paper.”

 „The authors should consider the possibility of a misdiagnosis of malignant skin disorders, including parameters of high-risk patients.”

- Lyundup, A.V.; Balyasin, M.V.; Maksimova, N.V.; Kovina, M.V.; Krasheninnikov, M.E.; Dyuzheva, T.G.; Yakovenko, S.A.; Appolonova, S.A.; Schiöth, H.B.; Klabukov I.D. Misdiagnosis of diabetic foot ulcer in patients with undiagnosed skin malignancies. International Wound Journal 2022, 19(4): 871-887. https://doi.org/10.1111/iwj.13688

In a reverse manner, the possibility of a misdiagnosis of malignant skin disorders, inlcuding non-melanoma skin cancer, is a main challenge in terms of worldwide public health management. A lot of clinicians may face it and thus malignancy-related misdiagnosis remains one of the main issues in the dermatologic field. Similar to the case of melanomas or basal cell carcinomas being misdiagnosed as diabetic foot ulcer, they can be easier considered, even by experts, benign lesion such as dermatofibromas [111]. As malignant tumors may sometimes mimic benign conditions, the main focus has been on finding a non-invasive, reliable, sensible and highly specific diagnostic methods to identify specific features which suggest malignancy.

Reference 111 (Lyundup et al) was inserted in the text.

Best Regards,

Beatrice Balaceanu, MD

Round 3

Reviewer 2 Report

The authors have satisfactorily responded to all my comments and made the necessary changes to the manuscript. However, I recommend collating separate figures: 14 figures in a single paper are too much.